# Noisy NER with Uncertainty-Guided Tree-Structured CRFs

**Jian Liu**[1]*, **Weichang Liu**[1]*, **Yufeng Chen**[1], **Jinan Xu**[1], **Zhe Zhao**[2]

[1] Beijing Key Lab of Traffic Data Analysis and Mining, Beijing Jiaotong University
[2] Tencent AI Lab

{jianliu, weichangliu, chenyf, jaxu}@bjtu.edu.cn
nlpzhezhao@tencent.com

## Abstract

Real-world named entity recognition (NER) datasets are notorious for their noisy nature, attributed to annotation errors, inconsistencies, and subjective interpretations. Such noises present a substantial challenge for traditional supervised learning methods. In this paper, we present a new and unified approach to tackle annotation noises for NER. Our method considers NER as a constituency tree parsing problem, utilizing a tree-structured Conditional Random Fields (CRFs) with uncertainty evaluation for integration. Through extensive experiments conducted on four real-world datasets, we demonstrate the effectiveness of our model in addressing both partial and incorrect annotation errors. Remarkably, our model exhibits superb performance even in extreme scenarios with 90% annotation noise.

## 1 Introduction

Named entity recognition (NER) is a fundamental natural language processing task that aims to find entities with certain types in texts (Ma and Hovy, 2016; Lample et al., 2016). Generally, building a high-performance NER model requires to obtain high-quality labeled data. However, due to the complexities of the labeling process, real-world NER datasets often contain annotation noises (Lan et al., 2020; Huang et al., 2021a). For example, Figure 1 gives two typical annotation errors in the CoNLL03 benchmark (Tjong Kim Sang and De Meulder, 2003) — the PERSON entity "Newcombe" is missed by annotators, and "Loc Angeles Lakers" is incorrectly labelled as LOCATION. Indeed, the noise rate is high (2%-8%) in real-world datasets (Liu and Tao, 2016; Song et al., 2022), and this issue can significantly impede model learning.

To date, many studies have been proposed to address NER annotation noises. For example, for the

---

*Equal contribution.

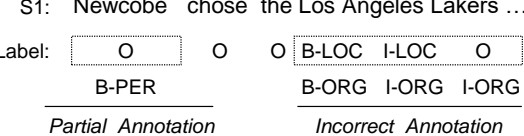

Figure 1: Two typical annotation noises, taken from the CoNLL 2003 NER benchmark.

partial annotation noise, previous works have employed re-weighting mechanisms to mitigate the impact of false negatives (Peng et al., 2019; Jie et al., 2019). For the incorrect annotation noise, previous works have suggested that the use of reinforcement learning and teacher-student framework is effective (Zhang et al., 2021; Zhou et al., 2022). Nevertheless, most works focus on only one type of noise, and there has been limited success in using a unified approach to handle diverse types of noises (Qu et al., 2022; Huang et al., 2021b). Additionally, due to the combination of different components, previous methods may suffer from error propagation (Peng et al., 2019; Wang et al., 2019b).

In this paper, we present a unified approach for addressing annotation noises in NER. Specifically, as shown in Figure 2, our method structures NER as a constituency tree parsing problem, with each node's label representing the entity type of a textual span. To learn with noises, we introduce uncertain nodes in the tree, indicating that the annotation for a span may be unreliable (this is compatible with both partial and incorrect annotation noises), and we use tree-structured conditional random fields (CRFs) (Rush, 2020) to average such uncertainties for learning, relieving the model from over-training on the noises. We present an Monte Carlo Dropout (MC-Dropout) mechanism (Gal and Ghahramani, 2016) for evaluating uncertainty and demonstrate that the overall framework is compatible with an iterative co-learning framework.

To verify the effectiveness of our method, we have conducted extensive experiments on real-world and simulated datasets. According to the results, on real-world datasets including Youku (Yang et al., 2020) and Weibo (Peng and Dredze, 2017), our approach outperforms previous state-of-the-art methods by up to 3.27% in absolute F1 score (§ 5.1), demonstrating remarkable effectiveness in addressing both partial and incorrect annotations. On simulated data sets, our method exhibits advantages under extreme conditions. For example, in the case of 90% partial or incorrect annotations on the TaoBao dataset (Jie et al., 2019), our approach achieves an impressive F1 score of 77.3%/25.36%, outperforming previous state-of-the-art methods by a margin of 10.4%/2.13% (§ 5.2 and § 5.3).

In summary, our contributions are as follows:

- We have introduced a new framework that can effectively handle NER annotation noises. This framework stands out for its unified approach in addressing partial and incorrect annotations.

- We have introduced tree-structured CRFs with uncertainty nodes for learning, along with the incorporation of MC-Dropout for uncertainty evaluation. This approach provides a fresh perspective on noise modeling that can be applied to other tasks.

- We have conducted extensive experiments on both real-world and simulated datasets, demonstrating promising results. To promote further exploration, we have made our code publicly available at `https://github.com/feili583/NER`.

## 2 Related Work

### 2.1 NER with Annotation Noises

The issue of addressing annotation noises in NER has received significant attention in the research community (Yang et al., 2018; Jie et al., 2019; Li et al., 2022). In the case of partial annotations, previous methods have explored various techniques, including the utilization of partial CRFs (Yang et al., 2018; Jie et al., 2019), binary classification approaches (Shang et al., 2018), and PU learning strategies (Mayhew et al., 2019; Peng et al., 2019). Recent works have introduced innovative strategies such as adaptive negative sampling to effectively prevent the inclusion of unlabeled noises

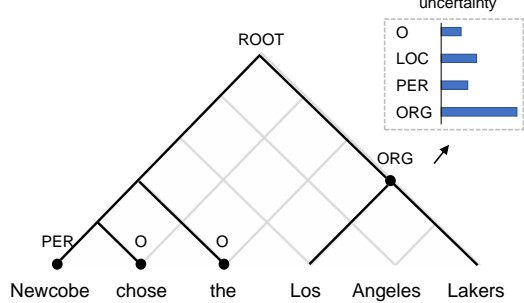

Figure 2: Tree-structured CRFs with uncertain/latent node. The uncertainty node means that the annotated label may not be reliable.

(Li et al., 2020, 2022). Regarding incorrect annotation noise, prior approaches have primarily focused on reinforcement learning methods (Yang et al., 2018; Luo et al., 2020; Chen et al., 2020), teacher-student frameworks (Wang et al., 2019b; Liang et al., 2020), and advanced noise filtering mechanisms (Zhang et al., 2021; Qu et al., 2022; Wang et al., 2023; Huang et al., 2021b). However, these methods are often tailored to address specific types of annotation noise, limiting their ability to generalize to different scenarios. In contrast, our research aims to propose a unified framework capable of effectively handling various types of annotation noises simultaneously, yielding a more comprehensive and versatile solution.

### 2.2 NER as Parsing

Our work also relates to research that takes a structured prediction perspective, particularly constitutional parsing, on NER (Finkel and Manning, 2009; Yu et al., 2020; Fu et al., 2021a; Gómez-Rodríguez and Vilares, 2018). For example, Wang et al. (2019a) improve Chinese NER by incorporating features from neural semi-CRFs and neural tree-CRFs. Fu et al. (2021b) apply parsing techniques to handle nested NER and introduce latent nodes for likelihood integration. Yang and Tu (2022) demonstrate the ability to help entity boundary detection by leveraging structure features. Motivated by these prior approaches, we structure NER as a constituency tree prediction problem. However, we extend this perspective by introducing uncertain nodes to effectively handle annotation noises. We hope that this new approach to addressing noises has the potential to inspire further research in other tasks.

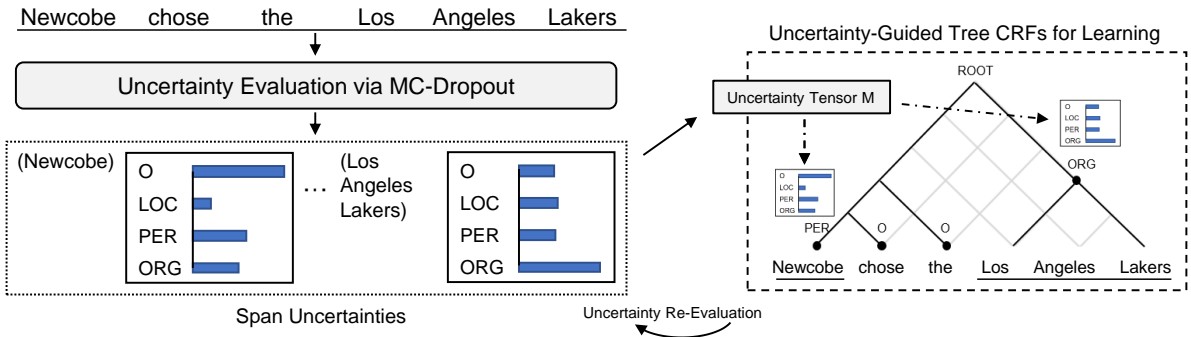

Figure 3: Overview of our approach. Specifically, we treat NER as a constituency tree parsing problem and utilize tree CRFs with uncertain nodes to handle noisy annotations (§ 3.1). The MC-Dropout is used for uncertainty evaluation (§ 3.2), and the above two modules form an iterative learning framework to improve each other (§ 3.3).

## 3 Approach

Our approach, depicted in Figure 3, consists of three main components:

- Uncertainty-guided tree CRFs. This module frames NER as a constituency tree parsing problem and utilizes tree conditional random fields (CRFs) with uncertain nodes to handle noisy annotations (§ 3.1).

- MC-Dropout for uncertainty evaluation. This module estimates the labeling uncertainty for a text span using MC-Dropout (Gal and Ghahramani, 2016), and integrates it into tree CRFs for learning with noises (§ 3.2).

- Iterative co-learning mechanism. This mechanism enables interactions between the above two modules, allowing them to learn from each other for improvement (§ 3.3).

### 3.1 Uncertainty-Guided Tree CRFs

The uncertainty-guided tree CRFs covert NER as constituency tree parsing and incorporate uncertainty mechanisms to address annotation noises. Let us consider a sentence $X$ with $N$ words: $X = \{x_1, x_2, .., x_N\}$. A constituency tree for the sentence can be represented as a rank-3 binary tensor $\boldsymbol{T} \in \mathbb{R}^{N \times N \times K}$, where the entry $\boldsymbol{T}_{i,j,k} = 1$ indicates that the span $[x_i, \ldots, x_j]$ is assigned an entity label $k$ from a label set $\mathbb{K}$ (denoted as $[i, j] \to k$). Let us assume that we can acquire a score tensor $\boldsymbol{S} \in \mathbb{R}^{N \times N \times K}$, where $\boldsymbol{S}_{i,j,k}$ represents the log potential for an assignment $[i, j] \to k$. Then the score[1] of the constituency tree $\boldsymbol{T}$ can be for-

---

[1]We assume a 1-order representation.

mulated as follows (Dozat and Manning, 2017):

$$\text{score}(\boldsymbol{T}, \boldsymbol{S}) = \sum\nolimits_{i,j,k} \boldsymbol{T}_{i,j,k} \boldsymbol{S}_{i,j,k} \qquad (1)$$

and the (logarithm) Gibbs distribution of $\boldsymbol{T}$ over all compatible constituency trees is:

$$\log p(\boldsymbol{T}|\boldsymbol{S}, X) = \text{score}(\boldsymbol{T}, \boldsymbol{S}) - \log Z \qquad (2)$$

Here $Z$ is a normalization term that sums scores of all constituency trees: $Z = \sum_{\hat{\boldsymbol{T}} \in \mathbb{T}} \exp \text{score}(\hat{\boldsymbol{T}})$, and in the binary constituency tree setting, we can compute it effectively using the Inside algorithm over a tree CRFs formulation (Eisner, 2016), denoted by $Z = \text{INSIDE}(\boldsymbol{S})$. In our approach, we compute each entry in the score tensor $\boldsymbol{S}$ using biaffine attentions (Dozat and Manning, 2017):

$$\boldsymbol{S}_{i,j,k} = \boldsymbol{H}_i^T \boldsymbol{W}_k^{(1)} \boldsymbol{H}_j + (\boldsymbol{H}_i + \boldsymbol{H}_j)^T \boldsymbol{W}_k^{(2)} \quad (3)$$

where $\boldsymbol{H}_i \in \mathbb{R}^d$ is the representation of word $x_i$ from a BERT encoder (Devlin et al., 2019), and $\boldsymbol{W}_k^{(1)}$, $\boldsymbol{W}_k^{(2)} \in \mathbb{R}^{d \times d}$ are parameters optimized through the training process. When we have access to the ground-truth constituency tree, denoted as $\boldsymbol{T}_X$, we can directly maximize $\log p(\boldsymbol{T}_X|X)$ for learning. For inference, given $\boldsymbol{S}$, we can apply the Cocke–Younger–Kasami (CKY) algorithm (Sakai, 1961) to find the most probable constituency tree with an O($N^3$) time complexity.

Nevertheless, the above formulation is applicable only when the clean constituency tree $\boldsymbol{T}_X$ is available. In our problem, we are confronted with noisy labels, which could result in mis-training. To address this challenge, we introduce uncertain nodes associated with the tree, represented by $\boldsymbol{U} \in \mathbb{R}^{N \times N \times K}$. Here, $\boldsymbol{U}_{i,j,k}$ indicates the uncertainty (or risk) of the model assigning label $k$ to the

span $[i, j]$ during the learning process. Different from previous methods learning a transition probabilities from correct labels to nosiy ones (Wang et al., 2019b), we propose a more principled approach by integrating uncertainties over all compatible constituency tree, and obtain an average score for a sentence $X$:

$$\text{score}(\boldsymbol{S}, \boldsymbol{U}) = \log \sum_{\hat{\boldsymbol{T}} \in \mathbb{T}} \exp(\sum_{i,j,k} \boldsymbol{U}_{i,j,k} S_{i,j,k})$$
(4)

Notice that the noisy labels are not used to compute this score, and in the sense that we integrate all constituency tree out, it resembles an unsupervised learning (The noisy labels are used to measure the uncertainties). For learning, we maximize:

$$\log p(S, X, U) = \text{score}(\boldsymbol{S}, \boldsymbol{U}) - \log Z \quad (5)$$

It should also note that directly computing $\text{score}(\boldsymbol{S}, \boldsymbol{U})$ is infeasible due to the exponential many terms. We show that it has a connection with the original inside algorithm by absorbing the uncertainty tensor into the score tensor and thus use the original Inside algorithm:

$$\text{score}(\boldsymbol{S}, \boldsymbol{U}) = \text{INSIDE}(\log \boldsymbol{U} \oplus \boldsymbol{S}) \quad (6)$$

where $\oplus$ is an element-wise add operator. Notice that the above process only applies at the training and does not affect the inference procedure.

### 3.2 Uncertainty Evaluation via MC-Dropout

This modules aims to measure the uncertainty tensor $\boldsymbol{U}$ using the original labels. Specifically, for an input sentence $X$, we first employ another BERT (Devlin et al., 2019) encoder to learn the representations $\boldsymbol{H} \in \mathbb{R}^{N \times d}$:

$$\boldsymbol{H} = \text{Encoder}(X) \quad (7)$$

Then for each span $[i, j]$, we build a representation $\boldsymbol{R}_{i,j} \in \mathbb{R}^{4d}$ following Li et al. (2020):

$$\boldsymbol{R}_{i,j} = \boldsymbol{H}_i \oplus \boldsymbol{H}_j \oplus (\boldsymbol{H}_i - \boldsymbol{H}_j) \oplus (\boldsymbol{H}_i \odot \boldsymbol{H}_j)$$
(8)

where $\oplus$ and $\odot$ are element-wise plus and multiplication operators. We then build a classifier to compute the probability for a span $[i, j]$ being each entity label $k \in \mathbb{K}$:

$$\boldsymbol{o}_{i,j} = \text{softmax}(\boldsymbol{W} \boldsymbol{R}_{i,j}) \quad (9)$$

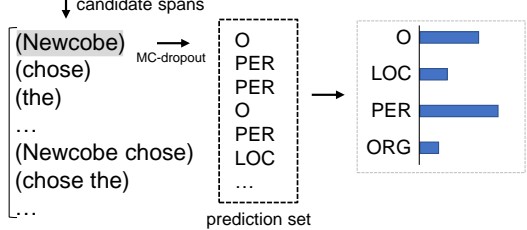

Figure 4: The process of uncertainty evaluation, using the first 1-word span "Newcobe" for illustration.

where $\boldsymbol{W} \in \mathbb{R}^{|\mathbb{K}| \times 4d}$ are model parameters.

It might be alluring to directly use $\boldsymbol{o}_{i,j}$ for constructing the uncertainty tensor, but according to previous research, the output of a network is often poorly calibrated, which can output very high probability even when the results is not reliable (Guo et al., 2017). Here motivated by Gal and Ghahramani (2016), we introduce MC-Dropout to evaluate the uncertainty. Specifically, after training the model using the original labeled data, we use it to predict the labels for $T$ times but with dropout layers activated. In this way, for a span, we can obtain a prediction set:

$$Y_{[i,j]} = [y_1, y_2, ..., y_T] \quad (10)$$

According to Gal and Ghahramani (2016), the variance of $Y_{[i,j]}$ indicates the uncertainty of the prediction. We therefore formulate it as a categorical distribution, and then for each entity label $K \in \mathbb{K}$, the entity probability parameter is:

$$c_{i,j,k} = \frac{\exp N(k)}{\sum'_k \exp N(k')} \quad (11)$$

with $N(k)$ being the frequency of seeing $k$ in the prediction set. Figure 4 showcases a running example for uncertainty estimation.

### 3.3 An Iterative Co-Learning Mechanism

We utilize an iterative co-learning mechanism where tree CRFs and uncertainty evaluation modules alternate as teacher and student roles.

**Step 1: Initial Uncertainty Estimation.** We estimate the initial uncertainty tensor $\boldsymbol{U}$ by training on the noisy labels from the training set, capturing the model's confidence in its predictions.

**Step 2: Training the Parsing Model.** The parsing model is trained by integrating the estimated uncertainties, adapting its learning process based on varying confidence levels associated with labeled data.

**Step 3: Uncertainty Re-Evaluation.** We refine uncertainty estimation by retraining the model using prediction results from the parsing model, enhancing its ability to estimate uncertainties effectively. We then alternate between Step 2 and Step 3 until convergence.

## 4 Experimental Setups

### 4.1 Datasets and Evaluations

We conduct experiments on four NER datasets: Youku (Yang et al., 2020), Weibo (Peng and Dredze, 2015, 2017), CoNLL-2003 (Wang et al., 2019b), and TaoBao (Jie et al., 2019). Our real-world evaluation settings are as follows: 1) For the Youku dataset, we directly adopt it for training and testing, which consists of a noisy training set with entity annotations and a clean test set. 2) For the Weibo dataset, we use the original noisy version for training and validation based on (Peng and Dredze, 2015). For testing, we utilize the corrected version mentioned in (Peng and Dredze, 2017). 3) In the case of the CoNLL03 dataset, we use the original training and validation sets from (Tjong Kim Sang and De Meulder, 2003). Additionally, we incorporate the corrected test set provided in (Wang et al., 2019b) for our evaluation.

In addition, we perform simulated experiments to assess the performance of our model under separate cases: partial annotations and incorrect annotations, as well as extreme conditions. Regarding the Weibo dataset, we divide the noisy training set into three parts[2], using the corrected version (Peng and Dredze, 2017) as a reference: clean set $S_{clean}$, partially annotated set $S_{partial}$ (accounting for 27.9%), and incorrectly annotated set $S_{incorrect}$ (accounting for 8.49%). We combine $S_{clean}$ and $S_{partial}$ to form a dataset containing only partial annotation noise, and we combine $S_{clean}$ and $S_{incorrect}$ to form a dataset containing only incorrect annotation noise. For the CoNLL03 and TaoBao datasets, which exhibit lower levels of noise, we adopt the following strategies to construct evaluation datasets: 1) Partial Annotation: randomly masking labels for a certain percentage of entities, and 2) Incorrect Annotation: randomly substituting a certain percentage of entity annotations with labels from other types.

---

[2]In the cases where a sentence contains two types of errors, we correct the partial annotations and categorize them as incorrect annotations.

### 4.2 Baselines for Comparison

We compare our approach with several state-of-the-art methods, including: BERT, which utilizes the original BERT model for NER, without considering any specific mechanism to address annotation noises. Weighted PA (Jie et al., 2019), which incorporates marginal probability to handle incomplete annotations. To ensure a fair comparison, we modify it by integrating a BERT representation. NegSampling (Li et al., 2020), which adopt negative sampling to relieve the problem of over-training on false negatives. Con-MPU (Zhou et al., 2022), which employs PU learning to handle partial annotations. BOND (Liang et al., 2020), which adapts self-training techniques to tackle distant supervision scenarios. NNCE (Liu et al., 2021), which identifies noisy annotations by using confidence thresholds. We also incorporate modifications to BERT representations for comparison purposes. SCDL (Zhang et al., 2021), which utilizes a teacher-student framework to select pseudo-labels. ATSEN (Qu et al., 2022), which augments the teacher model with an adaptive mechanism and student ensemble. Partial Tree (Fu et al., 2021b), which utilizes partially observed Tree CRFs to address nested NER.

### 4.3 Implementations

In our implementations, we use $BERT_{base}$ encoder (Devlin et al., 2019) as the basic feature extractor. For Tree CRFs, the learning rates on the Youku, Weibo, and CoNLL-2003 datasets are set to 1e-5, while on the TaoBao dataset, it is set to 3e-5. These values are chosen from the options: 1e-5, 5e-5 and 1e-4. The batch size is set to 48, and we train with a window context of 64 words. For the uncertainty evaluation step, we employ MC-Dropout with a different number: We set $T$=30 for the Youku dataset and $T$=10 for the other datasets. These values are determined through a grid search process. To control the training process, we set a maximum of 20 epochs and use a time decay learning rate strategy. This strategy ensures that the training terminates after reaching the maximum number of epochs.

## 5 Experimental Results

### 5.1 Evaluations on Real-World Datasets

We first conducted an evaluation of our method on real-world NER datasets that include natural annotation errors introduced by annotators (such as

| Approach | Youku | | | Weibo | | | CoNLL03 | | |
|---|---|---|---|---|---|---|---|---|---|
| | P | R | F1 | P | R | F1 | P | R | F1 |
| BERT | 73.81 | 29.71 | 42.37 | 52.45 | 36.06 | 42.74 | 90.54 | 91.14 | 90.84 |
| *Partial Annotation Settings* | | | | | | | | | |
| Weighted-PA (Jie et al., 2019) | 71.62 | 74.72 | 73.14 | 57.30 | 48.80 | 52.71 | 85.58 | 87.73 | 86.65 |
| NegSampling (Li et al., 2020) | 68.10 | 68.59 | 68.35 | 64.15 | 55.31 | 59.40 | 93.23 | 92.23 | 92.73 |
| Conf-MPU (Zhou et al., 2022) | 61.77 | 64.69 | 58.78 | 56.77 | 36.47 | 44.41 | 80.79 | 89.55 | 84.94 |
| *Incorrect Annotation Settings* | | | | | | | | | |
| BOND (Liang et al., 2020) | 70.64 | 33.61 | 45.55 | 59.24 | 49.03 | 53.65 | 92.00 | 92.63 | 92.31 |
| NNCE (Liu et al., 2021) | 71.48 | 74.21 | 72.82 | 49.02 | 59.81 | 53.88 | 80.99 | 85.40 | 83.14 |
| SCDL (Zhang et al., 2021) | 70.36 | 48.69 | 57.55 | 62.16 | 35.20 | 44.95 | 93.25 | 91.94 | 92.59 |
| ATSEN (Qu et al., 2022) | 71.47 | 52.74 | 60.69 | 59.36 | 38.01 | 46.35 | 93.32 | 92.27 | 92.79 |
| *Nested Entity Setting* | | | | | | | | | |
| Partial-Tree (Fu et al., 2021b) | 69.73 | 66.68 | 68.17 | 64.14 | 54.39 | 58.86 | 92.24 | 92.51 | 92.37 |
| Ours | 75.72 | 77.11 | **76.41** | 61.10 | 59.18 | **60.12** | 93.47 | 92.16 | **92.81** |

Table 1: Results on three real-world NER benchmarks, where the previous methods are categorized by the scenarios they were originally proposed for.

| Approach | P | R | F1 |
|---|---|---|---|
| BERT | 57.53 | 40.19 | 47.32 |
| Weighted-PA (2019) | 65.92 | 56.46 | 60.82 |
| NegSampling (2020) | 66.63 | 57.87 | 61.91 |
| Conf-MPU (2022) | 24.18 | 15.94 | 19.21 |
| BOND (2020) | 64.98 | 50.00 | 56.52 |
| NNCE (2021) | 49.13 | 60.53 | 54.23 |
| SCDL (2021) | 54.81 | 29.01 | 37.94 |
| ATSEN (2022) | 57.26 | 36.22 | 44.38 |
| Partial-Tree (2021b) | 72.52 | 47.08 | 57.09 |
| Ours | 65.97 | 60.87 | **63.32** |

Table 2: Results on Weibo dataset with partial annotation settings.

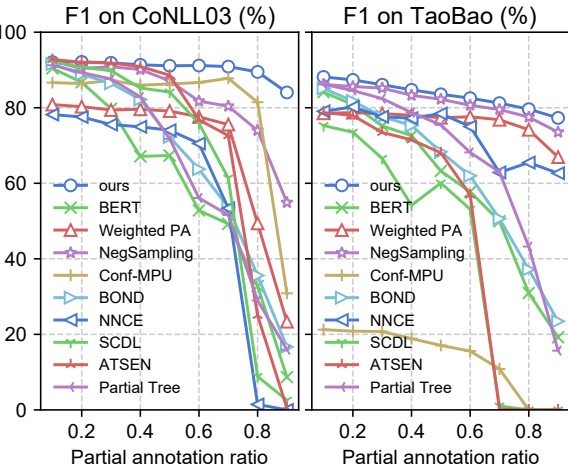

Figure 5: Results on CoNLL03 and TaoBao datasets regarding different partial annotation ratio. The performance is based on a 5-run average.

Weibo and CoNLL-2003) or machines (such as Youku). As illustrated in Table 1, our method outperforms all other approaches, achieving the highest performance. Notably, when compared to the traditional BERT model, our method exhibits substantial improvements in F1 score: 34.04% for Youku, 17.38% for Weibo, and 1.97% for CoNLL03. These results demonstrate the effectiveness of our method in practical use.

Moreover, our method outperforms strong baselines in handling both incomplete and incorrect annotations, demonstrating its ability to simultaneously resolve both categories of errors. It also exhibits improved applicability to datasets with a higher level of noises such as Youku and Weibo, resulting in an absolute F1 score enhancement of up to 3.27%. In addition, our method obtains a high recall rate, ranking first for Youku and second for Weibo, which demonstrates its robustness

in mitigating false negatives in partial annotations.

## 5.2 Evaluations on Partial Annotations

To gain a comprehensive understanding of our method's capability in handling partial annotations, we conducted more detailed experiments using simulated settings. Table 2 presents the results of our method on the Weibo dataset, where the training set consists solely of clean and partial annotations (§ 4.1). The results indicate a notable improvement of 1.41% in F1 score compared to the state-of-the-art approach. Moreover, upon closer examination, we observed that the improvement primarily stems from an improved recall.

Figure 5 shows results on Taobao and CoNLL03 with partial annotation settings (by masking a por-

| Approach | P | R | F1 |
|---|---|---|---|
| BERT | 52.25 | 55.50 | 53.83 |
| Weighted-PA (2019) | 62.38 | 46.41 | 53.22 |
| NegSampling (2020) | 57.46 | 63.29 | 60.23 |
| Conf-MPU (2022) | 21.28 | 16.73 | 18.73 |
| BOND (2020) | 59.01 | 66.75 | 62.64 |
| NNCE (2021) | 47.04 | 43.78 | 45.35 |
| SCDL (2021) | 53.95 | 39.95 | 45.91 |
| ATSEN (2022) | 57.53 | 43.88 | 49.78 |
| Partial-Tree (2021b) | 61.66 | 58.77 | 60.18 |
| Ours | 63.94 | 61.70 | **62.80** |

Table 3: Results on Weibo dataset with incorrect annotation settings.

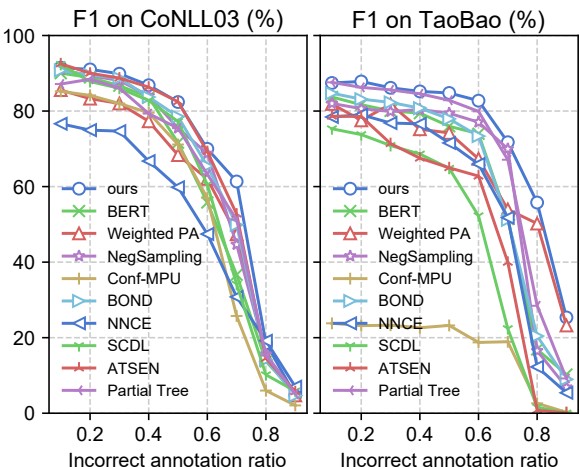

Figure 6: Results on CoNLL03 and TaoBao datasets regarding different incorrect annotation ratio. The performance is based on a 5-run average.

tion of entities), which demonstrate a notable margin between our approach and other methods. Remarkably, even when 90% of entities are masked, our method achieves an F1 score of over 80%, outperforming other methods significantly. This highlights the robustness of our approach in resisting partial annotation noises.

### 5.3 Evaluations on Incorrect Annotations

In addition, we conducted experiments utilizing simulate settings to verify the effectiveness of our method for addressing incorrect annotations. Table 3 shows results on Weibo, where the training set consists of only clean and incorrect annotations (§ 4.1). According to the results, our method outperforms or behaves similarly to earlier approaches, suggesting its effectiveness. Interestingly, we observed the highest precision in this incorrect annotation scenario, indicating that our approach automatically adopts a more cautious strat-

| Approach | P | R | F1 |
|---|---|---|---|
| Partial-Tree (2021b) | 64.14 | 54.39 | 58.86 |
| Softmax Probability | 62.30 | 56.28 | 59.14 |
| One Hot Prediction | 60.36 | 59.18 | 59.67 |
| Ours (MC-Dropout) | 61.10 | 59.18 | **60.12** |

Table 4: Ablations on uncertainty evaluation.

egy in scenarios involving incorrect annotations.

Figure 6 illustrates results for CoNLL 2003 and Taobao by substituting a percentage of entity labels with labels from other categories. According to the results, our method performs well in the vast majority of cases (especially on the TaoBao datasets), suggesting its effectiveness for resisting incorrect annotation noises. The generally degraded performance on CoNLL 2003 may be due to the dataset's limited entity coverage, and our stimulation strategy may result in numerous inconsistencies. In addition, we demonstrate that, compared to partial annotation, the advantage of our method diminishes in more challenging cases with higher incorrect annotation ratios (over 70% in CoNLL03 and 90% in TaoBao). Nevertheless, these extreme circumstances (such as a case that 90% annotations are wrong) are less common in real-world scenarios and require further study in future research.

## 6 Discussion

To further investigate the effectiveness of our approach, we conduct more in-depth studies, divided into ablation study and case study.

### 6.1 Ablation Study

**Effects of Uncertainty Evaluation.** Using the Weibo dataset as an example, we investigate the effects of uncertainty evaluation and compare our approach to Partial-Tree (Fu et al., 2021b), which treats O-label as latent nodes and does not adopt uncertainty evaluation, Softmax Probability, which uses the distribution as the uncertainty tensor, and One-Hot Prediction, which uses the argmax operation and constructs a one-hot uncertainty vector for a textual span. Based on the results in Table 4, the methods employing uncertainty evaluation outperform Partial-Tree, indicating that uncertainty evaluation is important in the noisy scenarios. Our method outperforms all others, suggesting that MC-Dropout is the most appropriate for evaluating uncertainty. One ex-

| Approach | P | R | F1 |
|---|---|---|---|
| One-Way Pass | 62.29 | 54.24 | 59.23 |
| BERT-BERT Co-Learning | 59.64 | 56.76 | 58.17 |
| Tree-Tree Co-Learning | 65.74 | 48.25 | 55.65 |
| Ours (BERT-Tree Co-Learn.) | 61.10 | 59.18 | **60.12** |

Table 5: Ablations on iterative co-learning mechanism.

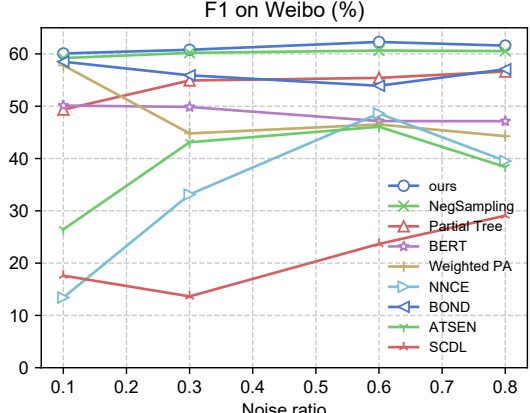

Figure 7: Effects of learning from pure noises.

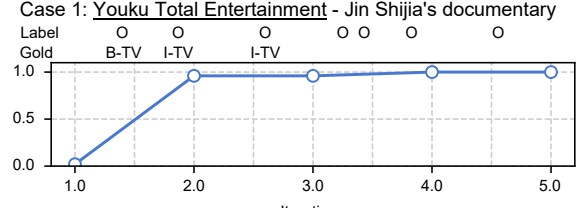

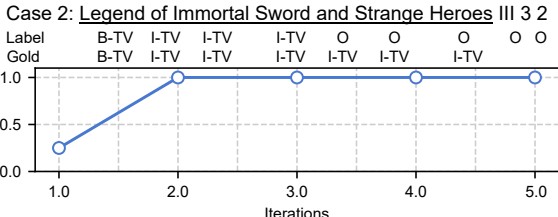

Figure 8: Two cases in the Youku dataset. It empha-sizes the progression of uncertainty evaluation in the gold annotation as the number of iterations increases. A higher value signifies a heightened confidence in the accuracy of the annotation. The model exclusively re-lies on the corresponding annotation as the gold label.

planation for the poor performance of Softmax Probability and One-Hot Prediction is that Soft-max Probability's output is typically poorly cal-ibrated and cannot accurately reflect uncertainty (Guo et al., 2017), whereas One-Hot Prediction is prone to error propagation.

**Effects of Iterative Co-Learning Mechanism.** To investigate the effects of the iterative co-learning mechanism, we compare our method to the following alternatives: One-Way Pass, which evaluates uncertainties only once and does not use iterative learning; BERT-BERT Co-Learning and Tree-Tree Co-Learning, which employ two BERT-based or tree-based models for uncertainty evalu-ation and NER learning. According to Table 5, One-Way Pass yields relatively bad results, indi-cating the significance of iterative co-learning. In addition, the results demonstrate that the BERT and tree CRFs have a complementary effect on un-certainty evaluation and training, and a mismatch will result in degraded performance.

**Effects of Learning from Pure Noises.** We conduct an interesting exploration to determine if a model can learn from pure noises by separat-ing all annotation errors (including partial and in-correct annotations) from the Weibo training set and progressively add them into clean datasets for learning. As demonstrated in Figure 7, our

method consistently outperforms others and shows continuous improvement as more noisy data is added. This indicates that the inclusion of pure noises is beneficial for learning, highlighting the advantages of our integrated training objective. In contrast, certain methods like BERT, BOND, and Weighted PA methods display a declining trend, implying that the addition of pure noises has a detrimental impact, leading to negative effects.

## 6.2 Case Study

In Figure 8, we present the evolution of uncer-tainty evaluation, represented by the span confi-dence of the ground-truth label, for two examples from the Youku dataset. Through the training pro-cess, our method significantly improves in accu-rately assigning correct labels to the initially mis-labeled entities. Moreover, the model's predic-tion confidence consistently increases over time, demonstrating its effective learning from noisy la-bels rather than relying solely on annotations.

## 7 Conclusion

This study introduces a novel and unified solu-tion to effectively address the noisy annotations in NER. Our model integrates tree CRFs with uncer-tainty evaluation, and comprehensive experiments validate the effectiveness of our approach in han-dling partial and incorrect annotations. Although our primary focus was NER, we anticipate that our method holds promise for broader applications in

various natural language processing tasks involving annotation noises.

## Acknowledgement

This work was supported by the Fundamental Research Funds for the Central Universities 2023JBMC058. It was also supported by the National Natural Science Foundation of China (No.62106016 and No.61976016), and the Open Projects Program of the State Key Laboratory of Multimodal Artificial Intelligence Systems.

## 8 Limitations

The study's findings should be considered in light of certain limitations. Firstly, it is important to recognize that our assumption of a domain-free formulation of noises may not fully account for domain-specific annotation errors. Indeed, real world annotation errors can vary across datasets and domains, emphasizing the need for future investigations that address domain-specific considerations. Secondly, a more promising approach to addressing these noises involves modeling the annotation process itself, rather than relying solely on learning from the noises. One potential solution is to develop an annotation model for each individual annotator to incorporate his/her annotation bias, despite the additional costs involved. In summary, exploring these complex issues and their potential applications holds promise for future research and requires further investigation.

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

# A  Dataset

We considered and conducted the evaluation of our methods on the following datasets. Basic data statistics are also reported in Table 7.

**Youku**  It is sourced from the NLPCC-2020 Shared Task on AutoIE and specifically involves caption text from YouKu videos (Yang et al., 2020). It focuses on three categories: TV, PER, and NUM. The data is divided into three sets for training, development, and testing. Training dataset is an unlabelled corpus containing 10,000 samples, and the entities are labelled by string matching with the given entity lists. Entity lists with specific category, which may cover around 30% of entities appearing in the unlabelled corpus. Dev dataset contains 1,000 samples with full label, and 2,000 samples with full label in test dataset.

**Weibo**  This dataset involves Chinese social media messages (Peng and Dredze, 2015). Annotations were initially generated using Amazon Mechanical Turk and subsequently refined by amalgamating labels from different Turkers, which introduced some inconsistencies and errors. The dataset was manually corrected by Peng and Dredze (2017), resulting in a cleaner version. It contains 1,890 messages sampled from Weibo between November 2013 and December 2014. The sentences in training set, dev set and test set is 1,350, 270 and 270. The entity types include Geopolitical, Location, Organization, and Person in both Name and Nominal forms. In the corrected version, partially annotated training set accounts

| | Data | Batch Size | LR | Train Time | Infer Time | Max Epochs | F1 ± std |
|---|---|---|---|---|---|---|---|
| BERT | Youku | 64 | 1e-4 | 00:11:11 | 00:00:05 | 20 | 41.50 ± 1.32 |
| | Weibo | 16 | 1e-4 | 00:03:47 | 00:00:02 | 20 | 44.62 ± 1.24 |
| | CoNLL03 | 64 | 1e-4 | 00:15:01 | 00:00:05 | 20 | 89.96 ± 0.52 |
| *Partial Annotation Settings* | | | | | | | |
| Weighted-PA (2019) | Youku | 32 | 1e-5 | 01:29:43 | 00:00:01 | 10 | 73.48 ± 0.32 |
| | Weibo | 32 | 1e-5 | 00:39:51 | 00:00:01 | 10 | 50.27 ± 1.54 |
| | CoNLL03 | 32 | 1e-5 | 06:11:53 | 00:00:04 | 10 | 82.98 ± 1.93 |
| NegSampling (2020) | Youku | 16 | 1e-5 | 00:43:28 | 00:00:03 | 40 | 67.89 ± 1.21 |
| | Weibo | 8 | 1e-5 | 00:19:18 | 00:00:05 | 40 | 60.00 ± 0.57 |
| | CoNLL03 | 16 | 1e-5 | 01:06:07 | 00:00:08 | 40 | 92.67 ± 0.08 |
| Conf-MPU (2022) | Youku | 128 | 5e-5 | 01:30:50 | 00:00:21 | 5 | 61.50 ± 4.90 |
| | Weibo | 16 | 5e-5 | 00:14:13 | 00:00:08 | 5 | 41.42 ± 2.51 |
| | CoNLL03 | 128 | 5e-5 | 02:15:40 | 00:08:47 | 5 | 84.07 ± 0.99 |
| *Incorrect Annotation Settings* | | | | | | | |
| BOND (2020) | Youku | 32 | 1e-5 | 00:58:07 | 00:00:01 | 50 | 45.20 ± 0.38 |
| | Weibo | 32 | 1e-5 | 00:06:36 | 00:00:01 | 50 | 54.14 ± 1.47 |
| | CoNLL03 | 128 | 1e-5 | 01:09:17 | 00:00:05 | 50 | 92.42 ± 0.14 |
| NNCE (2021) | Youku | 64 | 3e-5 | 03:27:15 | 00:00:01 | 10 | 73.34 ± 0.93 |
| | Weibo | 16 | 3e-5 | 02:16:54 | 00:00:01 | 10 | 56.38 ± 1.48 |
| | CoNLL03 | 64 | 1e-5 | 05:57:24 | 00:00:05 | 10 | 79.67 ± 1.84 |
| SCDL (2021) | Youku | 32 | 1e-4 | 03:26:52 | 00:00:01 | 50 | 57.97 ± 1.04 |
| | Weibo | 16 | 1e-4 | 00:25:20 | 00:00:01 | 50 | 37.53 ± 5.64 |
| | CoNLL03 | 32 | 1e-5 | 10:00:37 | 00:00:02 | 50 | 92.48 ± 0.12 |
| ATSEN (2022) | Youku | 8 | 1e-5 | 07:44:37 | 00:00:01 | 50 | 58.81 ± 1.04 |
| | Weibo | 16 | 1e-3 | 00:28:10 | 00:00:01 | 50 | 44.76 ± 1.66 |
| | CoNLL03 | 16 | 1e-5 | 16:05:15 | 00:00:39 | 50 | 92.83 ± 0.11 |
| *Nested Entity Setting* | | | | | | | |
| Partial-Tree (2021b) | Youku | 48 | 3e-5 | 01:16:41 | 00:00:07 | 30 | 70.13 ± 1.64 |
| | Weibo | 48 | 3e-5 | 00:29:27 | 00:00:03 | 30 | 58.43 ± 2.09 |
| | CoNLL03 | 96 | 3e-5 | 01:56:09 | 00:00:08 | 30 | 92.58 ± 0.23 |
| Ours | Youku | 48 | 3e-5 | 05:59:44 | 00:00:41 | 5 | 75.38 ± 1.14 |
| | Weibo | 48 | 3e-5 | 03:26:00 | 00:00:34 | 5 | 60.64 ± 0.78 |
| | CoNLL03 | 96 | 3e-5 | 09:41:59 | 00:02:30 | 5 | 92.85 ± 0.15 |

Table 6: Hyperparameters and computational time for baselines and our model (corresponding to the main results in Table 1). Note that for the "Max Epochs", certain models involve both inner and outer loops, and in this analysis, we've considered only the outer loops. Furthermore, the reported training time encompasses the cumulative time for all external loops, along with the inference time for a single iteration, expressed in hours, minutes and seconds.

for 27.9%, and incorrectly annotated training set makes up 8.49%.

**CoNLL03** This dataset comprises newswire from the Reuters RCV1 corpus (Wang et al., 2019b), tagged with four entity types (PER, LOC, ORG, MISC). It consists of standard training, development, and test sets. The statistics show sentence counts of 14,987, 3,466 and 3,684, although some records report 14,041, 3,250 and 3,453, as we did not exclude duplicates marked by '-DOCSTART-'. Wang et al. (2019b) has identified and corrected annotation mistakes in about 5.38% of test sentences.

**TaoBao** This dataset relates to the e-commerce domain and was crawled and manually annotated by Jie et al. (2019). It comprises sentences for training, development, and test sets with counts of 6,000, 998 and 1,000 respectively, covering four distinct entity types: PATTERN, PRODUCT, BRAND, and MISC.

## B  Models and Reproducibility

We provide average micro metrics over 5 seeds across each dataset in Table 1. And we present

| | Entity c(%) | Train | | Val | | Test | |
|---|---|---|---|---|---|---|---|
| | | entity | sent | entity | sent | entity | sent |
| Youku | 3 70.0 | 5,299 | 10,000 | 1,369 | 1,000 | 2,770 | 2,000 |
| Weibo | 8 36.4 | 1,370 | 1,350 | 301 | 270 | 414 | 270 |
| CoNLL03 | 4 05.4 | 23,499 | 14,987 | 5,942 | 3,466 | 5,702 | 3,684 |
| Taobao | 4 - | 29,397 | 6,000 | 4,941 | 998 | 4,866 | 1,000 |

Table 7: Data statisics. While Entity c(%) reveals the count of entity types along with their respective noisy ratios. Train, Validation and Test denote the quantities of entities and sentences within each dataset.

the standard deviation in Table 6. On baselines, where we do finet-tuning, some hyperparameters were mannually tuned but most left at their default values. The final values for the ones that were mannually tuned and the time for train and infer are also provided in Table 6.