# OpenReview forum: "Addressing NER Annotation Noises with Uncertainty-Guided Tree-Structured CRFs"
_EMNLP/2023/Conference — EMNLP 2023 Main_

### Official Review · Reviewer_hCvH · 2023-07-24

**Soundness:** 3

**Excitement:**

3: Ambivalent: It has merits (e.g., it reports state-of-the-art results, the idea is nice), but there are key weaknesses (e.g., it describes incremental work), and it can significantly benefit from another round of revision. However, I won't object to accepting it if my co-reviewers champion it.

**Missing References:**

[1]	 Lukas Lange et al., "Feature-dependent confusion matrices for low-resource ner labeling with noisy labels", in EMNLP-IJCNLP, 2019.

**Paper Topic And Main Contributions:**

This paper describes a method that focuses on training NER model on noisy annotations.
The authors frame the NER as a constituency parsing problem where each node in the constituency tree has a label representing the entity type of the textual span. To address annotation noise, the authors introduce uncertain nodes in the constituency tree. They use tree-structured CRFs to average uncertainty and MC-Dropout for evaluating uncertainty. Four NER datasets were used to evaluate the effectiveness of the proposed method. The experimental results show that the proposed method outperforms several baselines that focus on different types of noisy labels.

**Questions For The Authors:**

Question A: In section 3.2, the proposed method (MC-Dropout) seems to measure the uncertainty of model predictions rather than the reliability of the annotated label (Figure 2)? would you please clarify how these two concepts relate to each other?

Question B: Line 337: the way of creating incorrect annotation considers only entity type prediction error; suggest considering entity boundary errors as well, or explaining why boundary errors are not considered.

Question C: For MC-Dropout, different epochs are searched for MC-Dropout (line 337). Would you please discuss the impact of the number of epochs and clarify whether a similar grid search has been applied to other methods in Table 4?

Question D: Can you explain why Conf-MPU's performance on Taobao (Figure 5, 6) is much worse than other methods even when the ratio is 0, whereas, on other datasets, the gap between Conf-MPU and other methods seems to be smaller?

Question E: Line 506: not sure what 'pure noises' mean, especially when 'noise ratio' is also introduced in Figure 7?

**Reasons To Accept:**

A method of training NER model from imperfect annotations; empirical results show that the proposed method can effectively address different types of annotation errors.

**Reasons To Reject:**

There is a lack of clarity about where the main improvements come from (Table 4 and 5 show that the MC-Dropout and Co-learning are slightly better than alternatives); ~I also have a concern that the authors seem to report the best run results of their propose method but the baselines are not well-tuned~ (delete after rebuttal)

**Reproducibility:**

3: Could reproduce the results with some difficulty. The settings of parameters are underspecified or subjectively determined; the training/evaluation data are not widely available.

**Reviewer Confidence:**

4: Quite sure. I tried to check the important points carefully. It's unlikely, though conceivable, that I missed something that should affect my ratings.

**Typos Grammar Style And Presentation Improvements:**

Section 1 and Figure 1: suggest giving a brief definition of these two terms: `partial annotation` and `incorrect annotation`; because it seems that the example involving `Newcobe` is a missing (false negative) annotation; and the example involving `Los Angeles Lakers` can be attributed to the nested NER [1]; Arguably, `Los Angeles` is a partial annotation rather than an incorrect annotation.

Line 67: suggest adding the complete name (Monte Carlo?) when the abbreviation `MC' is first mentioned

Line 371: 1E-5 -> 1e-5, same changes applied to line 373

Suggest providing a summary table of used datasets, highlighting the different characteristics of noises (e.g., the source, noisy level). This may help the readers understand why very different patterns are observed on CoNLL03 and others.

Line 505: an interesting

Suggest adding efficiency analysis to understand the training and inference cost of the proposed method and alternatives.

[1]	 Nicky Ringland et al., "NNE: A Dataset for Nested Named Entity Recognition in English Newswire", in ACL, 2019.

---

> ### Author Rebuttal · Authors · 2023-08-29
>
> We value your insightful feedback and would like to provide a detailed response addressing each point.
>
> *For the weaknesses*
>
> **W1: A clarity about where the main improvements come from.**
>
> **A1**: Thank you for raising this concern. Our method can be thought of as an organically ensembled Tree-CRF model with an uncertainty evaluation module. To see where the main improvements come from, we refer to experiments in Tables 2 and 3, where, when compared to a vanilla Tree-CRFs, our method improves recall from 47.08 to 60.87 (+13.79%) on Weibo with partial annotations (Section 5.2), and from 58.77 to 61.70 (+2.93%) on Weibo with incorrect annotations (Section 5.3). The large improvement in partial annotations implies that by including uncertainty modeling within the CRFs, the model becomes more resistant to false negatives and resists their negative effects on training.
>
> Table 4 and 5 evaluate several uncertainty evaluation and co-training mechanism, but we believe that the new way to incorporate uncertainty assessment into a Tree-CRE parsing method is the most essential component for improvement. Results in Table 4 also support this claim. For example, even when augmenting vanilla CRFs with Softmax Probability (which is not a principled uncertainty measurement), the outcomes can be improved.
>
> We hope the above explanation clarifies the concern. Thank you.
>
>
>
> **W2: Regarding experimental settings.**
>
> **A2**: Thank you for expressing this concern. All of the results in our experiments are based on a 5-run average (as shown in the captions of Figures 5 and 6), and we tuned baselines using a grid search then perform the same 5-run average evaluation. As we discovered in our research, some baselines are unsuitable for real-world NER datasets. Conf-MPU, for example, uses external knowledge bases to generate noisy labels, but it is difficult to obtain related knowledge bases for real-world datasets like TaoBao, where the majority of entities are emergent names for items.
>
> In the revised version, we have provided the aforementioned explanation in the "Datasets and Evaluations" section, and we have also included standard deviation values of the F1 to the experiments, such as the 0.31 for the Weibo dataset (Table 1).
>
>
> *For the questions*
>
> **Q1: MC-Dropout seems to measure the uncertainty of model predictions rather than the reliability of the annotated label? would you please clarify how these two concepts relate to each other?**
>
> **A1**: We really appreciate this insightful question. We believe that the two concepts can be distinguished by using different modeling frameworks to explain them.
> -	For example, if we want to model "reliability of the annotated label", the best way is to use generative modeling and associate a latent variable to represent the ground-truth label. Then we should infer the label from the observed data, and the discrepancy between the inferred and observed labels represents the annotated label's reliability.
>
> -	In contrast, the goal of discriminative modeling methods (like neural network-based methods) is to directly predict the labels without requiring a ground-truth inferring procedure. In this way, we have to measure the uncertainty of the model itself, such as using the MC-Dropout method.
>
> Thanks.
>
>
> **Q2: Explaining why boundary errors are not considered.**
>
> **A2**: Thank you for this concern. In incorrect annotation simulations, we use span-level replacement and do not explore scenarios with boundary errors. One explanation is that entities with boundary errors are relatively rare in real-world annotations (occurring in fewer than 2% of cases in CoNLL 2003 and Weibo datasets), therefore an over-strength on such cases may make the evaluation unrealistic. The second reason is that we wish to separate partial annotation simulations from incorrect annotation simulations, however the boundary error may be a combination of two annotation issues, making a separated evaluation difficult.
>
>
> **Q3: Regarding giving the impact of the number of epochs in MC-Dropout and clarifying whether a similar grid search has been applied to other methods in Table 4?**
>
> **A3**: Thank you for this question. Do you mean the number of forward predictions (i.e., T in L268) when you refer "number of epochs in MC-Dropout"? Indeed, T is an MC-Dropout hyper-parameter; a large T makes the results more accurate but takes longer. In our approach, we use the smallest T value that is sufficient for uncertainty evaluation to achieve a speed-effectiveness balance (in fact, it is good to set T as 10 for Youku as well, and the results decrease about 0.7% in F1). However, none of the other methods in Table 4 have an equivalent hyper-parameter, thus we do not execute a grid search on them. Thanks.
>
>
> **Q4: Can you explain why Conf-MPU's performance on Taobao (Figure 5, 6) is much worse than other methods even when the ratio is 0, whereas, on other datasets, the gap between Conf-MPU and other methods seems to be smaller?**
>
> **A4**: Thank you for this question. One reason of the degraded performance of Conf-MPU is that it uses external knowledge bases to generate noisy labels, but it is difficult to obtain related knowledge bases for real-world datasets like TaoBao, where the majority of entities are emergent names for items
>
> **Q5: Line 506: not sure what 'pure noises' mean, especially when 'noise ratio' is also introduced in Figure 7?**
>
> **A5**: Thank you for this question. In "learning from pure noise", we aim to evaluate our model's ability to learn from *completely incorrect annotations (i.e., "pure noise")*. Assume we have a clean dataset and a second set of examples with completely incorrect entity labels. Then, it's common to believe that adding such incorrect examples to the clean dataset will impede learning and degrade performance. However, according to the experimental results, our model performs better when "pure noise" is added. This suggests that our model can learn from even pure noise, and that samples with completely incorrect labels can provide some supervision for learning.
>
> **Regarding typos, grammar style, and presentation improvements.**
>
> We highly appreciate the suggestions for revisions of typos, language style, and presentations and have incorporated them into the revised edition. In particular, in Figure 1, we have added a footnote explaining the incorrect annotation and its connection to the nested NER. We fixed typos and added a time comparison to compare the effectiveness of different methods. We have also included the missing reference for completeness.
>
> Thanks.

---

### Official Review · Reviewer_TyYs · 2023-08-03

**Soundness:** 3

**Excitement:**

4: Strong: This paper deepens the understanding of some phenomenon or lowers the barriers to an existing research direction.

**Paper Topic And Main Contributions:**

The paper presents an approach for training a noise-robust NER model that can learn well even in scenarios with extremely high noise rates in the training data. It consists of mainly two parts: Learning to assess uncertainty of the ground truth labels (via variance of predictions of different models when using MC-Dropout) and then training a tree-structured Conditional Random Fields (CRFs) that includes these uncertainty nodes. They apply an iterative co-learning mechanism with alternating between running the Uncertainty Evaluation and training the Tree CRFs parsing model, which helps the model to more and more train well on the real data and get better at detecting noise.

The authors present experiments on three real-word NER datasets where a noise-cleaned test set is available (Youku, Weibo, CoNLL03). Their method outperforms the BERT baseline as well as several other noise-robust models.

They also present experimental results on synthetic variants of these dataset where they differentiate between partial annotation errors (some entities are missing a label) and incorrect annotation errors (wrong label) and simulate extreme noise scenarios, showing that the model is especially robust when trained on partial annotations (meaning some labels are simply removed) compared to the other models. Interestingly (and maybe not discussed enough) when trained on high rates of incorrect annotation (labels are switched) the model shows surprisingly weaker quality (but still beating the rest of models).

Some ablation experiments are presented as well, showing the necessity of uncertainty evaluation and their co-learning mechanism.


**Questions For The Authors:**

see some suggestions made above (Reasons to Reject)

**Reasons To Accept:**

Annotation noise is a very important problem for NER and the paper presents an interesting approach to training noise-robust NER models. It is overall well written, structured and easy to follow the argumentation. The idea of alternating between uncertainty evaluation and model updates is inspiring and well described.
I appreciate especially that they used real-world datasets that happen to exist with noisy and clean variants/splits instead of just simulating noise artificially! But of course, also the synthetic experiments are very interesting and striking.
The experimental results are well presented and discussed, I appreciate the large number of competitive models that the authors compare their methods to.

**Reasons To Reject:**

I do not have major concerns about the paper. One thing that could be made clearer is that, when e.g. categorizing and simulation error types, they operate on token or span level (i.e. when adding synthetic partial errors: do whole mentions get assigned the O label or only random tokens?).
It seems that in the results tables (in contrast to those presented by the figures), they only present one specific model run. Given that those numbers do not show that much of deviations, it might be better to add mean and standard deviation of several runs.
The ablation experiments could benefit from a bit more care in their description, especially the one about "Learning from Pure Noise" which in my opinion was a bit hard to understand. E.g. make more clear what makes the "Pure Noise" experiments different to the experiments before. Also Sec. 6.2 Case Study and its figure should be described more clearly.

**Reproducibility:**

2: Would be hard pressed to reproduce the results. The contribution depends on data that are simply not available outside the author's institution or consortium; not enough details are provided.

**Reviewer Confidence:**

3: Pretty sure, but there's a chance I missed something. Although I have a good feel for this area in general, I did not carefully check the paper's details, e.g., the math, experimental design, or novelty.

**Typos Grammar Style And Presentation Improvements:**

* page 1, line 032: "Loc Angeles Lackers" > "Los Angeles Lakers"
* page 2, line 149: "we structures" > "we structure"
* page 4, line 263: check grammar
* Figure 8: "Glod" > "Gold"

---

> ### Author Rebuttal · Authors · 2023-08-29
>
> We sincerely appreciate your thoughtful feedback and would like to respond in detail.
>
> **Q1. When simulating error annotations, do you operate on token or span level?**
>
> **A1**: Thank you for expressing concern. We take into account *span level* in both partial annotation simulations and incorrect annotation simulations (such as masking an entire entity for partial annotation evaluation and replacing an entity's label for incorrect annotation evaluation), which guarantees a pure simulation evaluation scenario.
> In contrast, word-level masking/replacing may result in a mix situation (such as the Lincoln Memorial (LOC) being mistakenly annotated as Lincoln (PER)), making independent evaluation difficult. Moreover, given that such mixed cases are infrequent in real-world annotations (occurring in less than 2% of cases in CoNLL 2003 and Weibo datasets), we have opted for span-level simulation instead of word-level simulation.
>
>
> **Q2: Regarding the experimental setting and adding mean and standard deviation.**
>
> **A2**: Thank you for expressing concern. All of the results in our experiments are based on a 5-run average (as shown in the captions of Figures 5 and 6). We made it clearer in the new version by including this explanation in the "Datasets and Evaluations" section. In the upated we've also included standard deviation values of the F1 to the experiments, such as the 0.31 for the Weibo dataset (Table 1).
>
>
> **Q3: What is the goal of designing "learning from pure noise" experiments?**
>
> **A3**: Thank you for this question. The goal of designing "learning from pure noise" is to evaluate our model's ability to learn from *completely incorrect annotations (i.e., "pure noise")*. Assume we have a clean dataset and a second set of examples with completely incorrect entity labels. Then, it's common to believe that adding such incorrect examples to the clean dataset will impede learning and degrade performance. However, according to the experimental results, our model performs better when "pure noise" is added. This suggests that our model can learn from even pure noise, and that samples with completely incorrect labels can provide some supervision for learning.
>
> Lastly, we highly value the suggestions for typos, grammar, style, and presentation improvements and have incorporated them into the updated version. Thanks.

---

### Official Review · Reviewer_LrZJ · 2023-08-05

**Soundness:** 4

**Excitement:**

4: Strong: This paper deepens the understanding of some phenomenon or lowers the barriers to an existing research direction.

**Paper Topic And Main Contributions:**

This work proposes a new method for learning a NER tagger from very noisy training datasets, that either mislabel some entity mentions or fail to detect them. The authors demonstrate how to combine uncertainty estimation scores into a constituency parser that labels NER constituents. The method consists of two main components: a tree-based CRF for the parser, and a dropout-based uncertainty estimator.
These are trained using an iterative approach where each component's output serves to train the other in an iterative fashion.

**Reasons To Accept:**

- Very elegant modeling approach that makes clever use of classical results and tools from ML and NLP.
- Results demonstrate good performance on high-quality test sets as well as robustness to varying degrees of noise in the training set.
The authors add a thorough analysis on 4 different datasets as well as simulated data with different mislabeling conditions. The analysis also includes an ablation study for each component.
- The manuscript reads well and is well organized.

**Reasons To Reject:**

I thank the authors for addressing these concerns in their response:

- While the performance numbers are convincing, there is no error analysis of the results, and no way to interpret what the method does beyond looking at F1 scores.

- The reporting in the experimental design section could be improved, the datasets used include details such as domain, size, estimation of noise in the dataset, distribution of NER labels, and most importantly, how was the high-quality version of each test set obtained and what is its size.

**Reproducibility:**

4: Could mostly reproduce the results, but there may be some variation because of sample variance or minor variations in their interpretation of the protocol or method.

**Reviewer Confidence:**

4: Quite sure. I tried to check the important points carefully. It's unlikely, though conceivable, that I missed something that should affect my ratings.

---

> ### Author Rebuttal · Authors · 2023-08-29
>
> We really appreciate your feedback and would like to provide a point-by-point response.
>
> **W1: Regarding error analysis of the results and ways to interpret what the method does beyond looking at F1 scores.**
>
> **Q1**: Thank you for your concern, which will be especially helpful in enhancing the manuscript. Indeed, in addition to the F1 scores, we studied additional metrics to explain the strengths of our method. Refer to Line 405, where we discover that our method is particularly good at recall, implying that it can mitigate the negative effect of mislabeled examples (“In addition, our method obtains a high recall rate, ranking first for Youku and second for Weibo, which demonstrates its robustness in mitigating false negatives in partial annotations”).
>
> This claim was clarified in our revised version by using a more thorough comparison. Our method, in particular, raises the recall of a vanilla Tree-CRFs from 47.08 to 60.87 (+13.79%) on Weibo with partial annotations (Section 5.2), and from 58.77 to 61.70 (+2.93%) on Weibo with incorrect annotations (Section 5.3). The significant improvement in partial annotations shows that by incorporating uncertainty modeling within the CRFs, the model becomes more resistant to false negatives in annotations and resists its deleterious effects on training.
>
> We've also added a new error analysis section to compare the results of our method and a standard Tree-CRFs. Interestingly, the entities found by our methods convers most of the entities found by vanilla Tree-CRFs (96.4%), and in 67.4% of the extra entities discovered by our method, we can identify an unannotated example (false negative) with a similar pattern in the training set (judge by human). This is another evidence that we can reduce the impact of false negatives in training.
>
>
> **W2: Regarding adding more details of the datasets.**
>
> **Q2**: Thank you for this insightful suggestion. In the appendix, we have added further explanations about the datasets, which can be summarized as follows:
>
> -	YouKu Dataset: This dataset is sourced from the NLPCC-2020 Shared Task on AutoIE and specifically involves caption text from YouKu videos. It focuses on three categories: TV, person, and series. The data is divided into three sets for training, development, and testing. Train dataset: Unlabelled corpus containing 10000 samples, the entities are labelled by string matching with the given entity lists. Entity lists with specific category, which may cover around 30% of entities appearing in the unlabelled corpus. Dev dataset: 1000 samples with full label. Test dataset: 2000 samples with full label.
>
> -	Weibo Dataset: This dataset involves Chinese social media messages and annotations. The annotations were created using Amazon Mechanical Turk and later refined by merging labels from different Turkers. This process led to some inconsistencies and errors. The dataset was manually corrected over a period of two years, resulting in a cleaner version. It contains 1,890 messages sampled from Weibo between November 2013 and December 2014. The sentences in training set, dev set and test set is 1350,270 and 270. The entity types include Geo-political, Location, Organization, and Person. In the corrected version, partially annotated training set accounts for 27.9%, and incorrectly annotated training set accounts for 8.49%.
>
> -	CoNLL03 Dataset: This dataset comprises newswire from the Reuters RCV1 corpus, tagged with four entity types (PER, LOC, ORG, MISC). It consists of standard training, development, and test sets. The counts of sentences in these sets are 14,987, 3,466, and 3,684 respectively. There might be some statistics as 14041, 3250, 3453, because we didn't exclude the duplicates '-DOCSTART-'. Some papers identify and correct label mistakes in about 5.38% of test sentences.
>
> -	TaoBao Dataset: This dataset is related to the e-commerce domain and is crawled and manually annotated. It includes sentences for training, development, and test sets with counts of 6,000, 998, and 1,000 respectively. The dataset encompasses four types: PATTERN, PRODUCT, BRAND, and MISC.
>
> Thank you again.

---

### Meta-Review · Area_Chair_gpU9 · 2023-09-19

**Recommendation:** 4

**Metareview:**

The reviews have a consensus on positive evaluations for both soundness and excitement.

Regarding excitement, the proposed method of training an NER model from noisy data is novel.  In the line of framing NER as constituency parsing, the method introduces uncertain nodes in the constituency tree and trains tree-structured CRFs with MC-Dropout to measure uncertainty.

As for soundness, the paper provides extensive comparisons with several baselines on 4 datasets and their noisy variants with respect to 3 annotation error types.  More descriptions on the quality of test sets (e.g., the corrections that were made) would improve the soundness of evaluation.

---

### Decision · Program_Chairs · 2023-10-07

**Decision:**

Accept-Main

**Comment:**

The reviews have a consensus on positive evaluations for both soundness and excitement.

Regarding excitement, the proposed method of training an NER model from noisy data is novel.  In the line of framing NER as constituency parsing, the method introduces uncertain nodes in the constituency tree and trains tree-structured CRFs with MC-Dropout to measure uncertainty.

As for soundness, the paper provides extensive comparisons with several baselines on 4 datasets and their noisy variants with respect to 3 annotation error types.  More descriptions on the quality of test sets (e.g., the corrections that were made) would improve the soundness of evaluation.